Real-time reverse transcription polymerase chain reaction development for rapid detection of Tomato brown rugose fruit virus and comparison with other techniques

Panno Stefano 1 2 pannostefano@virgilio.it
Ruiz-Ruiz Susana 3
Caruso Andrea Giovanni 1
Alfaro-Fernandez Ana 4
Font San Ambrosio Maria Isabel 4
http://orcid.org/0000-0002-5926-6300 Davino Salvatore 1 5 salvatore.davino@unipa.it
1 Department of Agricultural, Food and Forest Sciences, University of Palermo , Palermo , Italy
2 Molecular Dynamics srl , RAGUSA , Italy
3 Fundación para el Fomento de la Investigación Sanitaria y Biomédica de la Comunitat Valenciana (FISABIO)-Salud Pública , Valencia , Spain
4 Instituto Agroforestal Mediteráneo, Universitat Politécnica de València (IAM-UPV) , Valencia , Spain
5 Institute for Sustainable Plant Protection, National Research Council (IPSP-CNR) , Turin , Italy
Lahr Daniel
Electronic publication date: 2019 Oct 17
Publication date: 2019
Volume: 7
Electronic Location ID: e7928
Received 2019 Aug 8; Accepted 2019 Sep 20
Copyright: © 2019 Panno et al.
Copyright year: 2019
Copyright holder: Panno et al.
License: This is an open access article distributed under the terms of the Creative Commons Attribution License, which permits unrestricted use, distribution, reproduction and adaptation in any medium and for any purpose provided that it is properly attributed. For attribution, the original author(s), title, publication source (PeerJ) and either DOI or URL of the article must be cited.
License URL: https://creativecommons.org/licenses/by/4.0/

Keywords: ToBRFV, RT-qPRC, Fast detection

Funding: The authors received no funding for this work.

==============================
Background

Tomato brown rugose fruit virus (ToBRFV) is a highly infectious tobamovirus that causes severe disease in tomato (Solanum lycopersicum L.) crops. In Italy, the first ToBRFV outbreak occurred in 2018 in several provinces of the Sicily region. ToBRFV outbreak represents a serious threat for tomato crops in Italy and the Mediterranean Basin.

Methods

Molecular and biological characterisation of the Sicilian ToBRFV ToB-SIC01/19 isolate was performed, and a sensitive and specific Real-time RT-PCR TaqMan minor groove binder probe method was developed to detect ToBRFV in infected plants and seeds. Moreover, four different sample preparation procedures (immunocapture, total RNA extraction, direct crude extract and leaf-disk crude extract) were evaluated.

Results

The Sicilian isolate ToB-SIC01/19 (6,391 nt) showed a strong sequence identity with the isolates TBRFV-P12-3H and TBRFV-P12-3G from Germany, Tom1-Jo from Jordan and TBRFV-IL from Israel. The ToB-SIC01/19 isolate was successfully transmitted by mechanical inoculations in S. lycopersicum L. and Capsicum annuum L., but no transmission occurred in S. melongena L. The developed real-time RT-PCR, based on the use of a primer set designed on conserved sequences in the open reading frames3, enabled a reliable quantitative detection. This method allowed clear discrimination of ToBRFV from other viruses belonging to the genus Tobamovirus, minimising false-negative results. Using immunocapture and total RNA extraction procedures, the real-time RT-PCR and end-point RT-PCR gave the same comparable results. Using direct crude extracts and leaf-disk crude extracts, the end-point RT-PCR was unable to provide a reliable result. This developed highly specific and sensitive real-time RT-PCR assay will be a particularly valuable tool for early ToBRFV diagnosis, optimising procedures in terms of costs and time.

Introduction

Tomato (Solanum lycopersicum L.) is one of the most important horticultural crops worldwide. Indeed in 2017, more than 182 million tons of tomato have been produced (FAO, 2017). China is the most important tomato producer (59 million tons), followed by India, Turkey and United States, while Italy and Spain are the major tomato producers in Europe (over six million and five million tons, respectively) (FAO, 2017). In Italy, greenhouse tomato production is mainly developed in the southern regions, and Sicily produces about 40% of national production (Agri ISTAT, 2017). In the last years, emerging viral diseases have been an important limiting factor for many crop production systems, such as tomato crops, causing considerable economic losses (Hanssen, Lapidot & Thomma, 2010). Greenhouse tomato production in Italy is affected by important losses that are caused by several different viruses: Pepino mosaic virus (Davino et al., 2017b, 2008; Tiberini et al., 2011), Tomato mosaic virus (ToMV), Tomato spotted wilt virus (Panno et al., 2012), Tomato leaf curl New Delhi virus (Panno et al., 2019), Tomato yellow leaf curl virus—Tomato yellow leaf curl Sardinia virus and their recombinants (Davino et al., 2009, 2012; Panno, Caruso & Davino, 2018) and Tomato brown rugose fruit virus (ToBRFV) detected at the end of 2018 (Panno, Caruso & Davino, 2019).

The international trade globalisation and the free commodities movement complicate the control of pathogens in the free trade area of the Mediterranean Basin. In this context, the control of seed-transmitted pathogens is extremely difficult. This is the case of the recent ToBRFV outbreak in Sicily, which was probably introduced into the island either through infected seeds or through infected fruits and their subsequent manipulation.

Tomato brown rugose fruit virus is a member of the genus Tobamovirus, family Virgaviridae. Tobamovirus is probably the largest genus of this family for numbers of species (King et al., 2011). Tobamoviruses are the only members of this family that have an undivided genome. ToBRFV has a single-stranded positive-sense RNA (gRNA) of ~6,400 nucleotides (nt), with a typical tobamoviruses organisation that consists in four open reading frames (ORFs) encoding two replication-related proteins of 126 and 183 kDa, in which the second protein is expressed by the partial suppression of the stop codon (ORF1 and ORF2), the movement protein (MP) of 30 kDa (ORF3) and the coat protein of 17.5 kDa (ORF4), which are expressed via the 3′-coterminal sub-genomic RNAs (Salem et al., 2016).

Tomato brown rugose fruit virus was described in 2016 for the first time in Jordan by Salem et al. (2016) on greenhouse tomato plants, and in 2017 in Israel on tomato plants harbouring the Tm-22 gene (Luria et al., 2017). Afterwards, ToBRFV was detected in Mexico in tomato and pepper crops (Cambrón-Crisantos et al., 2018) and after it spread to Germany, United States (California), Palestine, Italy and Turkey (Menzel et al., 2019; Ling et al., 2019; Alkowni, Alabdallah & Fadda, 2019; Panno, Caruso & Davino, 2019; Fidan, Sarikaya & Calis, 2019).

Tomato brown rugose fruit virus symptoms are typical of the tobamoviruses infection and consist of tomato leaves interveinal yellowing and deformation, mosaic staining, young leaves deformation and necrosis, sepal necrosis and deformation, young fruits discolouration, deformation and necrosis. For this reason ToBRFV represents a very dangerous problem for tomato crops in all the regions where tomato is cultivated, due to the ability of the virus to be transmitted by contact through contaminated tools, hands, clothing, direct plant-to-plant contact, propagation material (grafts, cuttings), bumblebees and seeds (Levitzky et al., 2019).

The aim of the present study was to characterise the virus found in Italy and to develop a sensitive, specific and economical method to detect ToBRFV in infected plants and seeds.

Materials and Methods

Source of viral material

In October 2018, virus-like symptoms typical of tobamovirus infection were observed in tomato greenhouses in the Ragusa province (Sicily, Italy). These symptoms were very similar to those described by Salem et al. (2016) who identify a new tobamovirus, named ToBRFV, which overcomes the Tm-22 resistance gene. The symptoms consisted of severe mosaic, young leaves deformation and necrosis, fruits discolouration and marbling and sepals’ necrosis (Fig. 1).

Figure 1 Symptoms of Tomato brown rugose fruits virus.

Severe mosaic (A), deformation and necrosis on young leaves (B); discolouration and marbling on fruits (C); necrosis on sepals (D).

The samples used in this study were collected in four different areas of Sicily, in the provinces of Agrigento, Caltanissetta, Ragusa and Siracusa. Fifteen samples were collected from two different greenhouses of each area, according to the following scheme: three plants rows were selected from 60 total rows (one row every 20) and five samples were taken from each selected row (one sample every 10 plants) (Panno, Caruso & Davino, 2019). Sampling was repeated three times: the first sampling in October 2018, the second in December 2018 and the third at the end of February 2019. A total of 360 samples were collected and marked by GPS using PLANTHOLOGY mobile application (Davino et al., 2017a).

Screening of ToBRFV using end-point RT-PCR

One-step end-point RT-PCR was performed in 25 μl (final volume) containing two μl of total RNA extract, 20 mM Tris–HCl (pH 8.4), 50 mM KCl, three mM MgCl2, 0.4 mM dNTPs, one mM of the forward primer ToBRFV-F-5722 and one mM of the reverse primer ToBRFV-R-6179 (Panno, Caruso & Davino, 2019), 4U of RNaseOut, 20U of superscript II reverse transcriptase-RNaseH and 2U of Taq DNA polymerase (Thermo Fisher Scientific, Waltham, MA, USA). RT-PCR was carried out in a MultiGene OptiMax thermal cycler (Labnet International Inc, Edison, NJ, USA) according to the following cycling conditions: 42 °C for 45 min, 95 °C for 5 min; 40 cycles of 30 s at 95 °C, 30 s at 55 °C, 30 s at 72 °C and a final elongation of 10 min at 72 °C. The obtained DNA products of expected size were confirmed by electrophoretic separation in a 1.5% agarose gel and staining with Sybrsafe (Thermo Fisher Scientific, Waltham, MA, USA).

Mechanical transmission

The Sicilian ToBRFV ToB-SIC01/19 isolate was mechanically inoculated into three plants of different hosts (S. lycopersicum L., S. melongena L. and Capsicum annuum L.) for subsequent biological characterisation. Plants were grown on a sterilised soil in an insect-proof glasshouse, with a photoperiod of 14 h of light and a target air temperature set at 28/20 °C day/night. The symptoms were reported weekly and the presence of ToBRFV was evaluated at 30 dpi by RT-PCR as previously described.

Full genome sequencing

Full genome sequencing was performed using the primer walking strategy (Knippers & Alpert, 1999) with the specific and overlapping primers reported by Luria et al. (2017). Four pairs of primers targeting ToBRFV conserved sequences were included to obtain the full genome sequence: FTobGEN, RTobGEN, F1-R1572, F1534-R3733 and F4587-R6392.

RT-PCR was performed in a MultiGene OptiMax thermal cycler (Labnet International Inc, Edison, NJ, USA) according to the conditions reported by Luria et al. (2017). The obtained products were confirmed by electrophoretic separation in a 1.5% agarose gel and visualised using Sybrsafe staining. The products were subsequently purified with the Ultraclean 96 PCR Cleanup Kit (Qiagen, Hilden, Germany) according to the manufacturer’s instructions and cloned using the TA-cloning system (Promega, Madison, WI, USA). The five obtained plasmids were sequenced in both directions using an ABI PRISM 3100 DNA sequence analyser (Applied Biosystems, Foster City, CA, USA).

To obtain the 3′ and 5′ terminal region sequences, a rapid amplification of cDNAs RACE was performed with the following primers: R-Ex-480, R-In-408, F-Ex-5931 and F-In-6041 (Luria et al., 2017), using a SMARTer® RACE 5′/3′ Kit (Takara Bio, Mountain View, CA, USA) according to the manufacturer’s instructions. The obtained DNA products were cloned using the TA-cloning system (Promega, Madison, WI, USA) and the five obtained plasmids were sequenced. All the obtained sequences of each region were assembled using the programme Contig implemented in Vector NTI Advance 11.5 software (Invitrogen, Carlsbad, CA, USA) to obtain the de novo full-length sequence of ToBRFV genome.

Primer and Taqman® MGB probe design

Four full-length genomic sequences of ToBRFV retrieved from GenBank (accession nos. MK133095, MK133093, KT383474 and KX619418) and a sequence assembled in our lab during this work were aligned using ClustalX2 programme (Larkin et al., 2007) in order to design five primer pairs targeting only conserved sequences.

The obtained primers were tested in vitro with the Primer-BLAST algorithm (https://www.ncbi.nlm.nih.gov/tools/primer-blast/index.cgi) to evaluate the possibility of specific hybridisation with other organisms. The same primers were also tested using Vector NTI Advance 11.5 software (Invitrogen, Carlsbad, CA, USA) with the complete sequence of other tobamoviruses to understand their affinity percentages. The included tobamoviruses were: Bell pepper mottle virus (one sequence), Brugmansia mild mottle virus (one sequence), Obuda pepper virus (two sequences), Paprika mild mottle virus (PaMMV) (two sequences), Pepper mild mottle virus (PMMV) (13 sequences), Rehmannia mosaic virus (six sequences), Tobacco mild green mosaic virus (TMGMV) (seven sequences), Tobacco mosaic virus (TMV) (10 sequences), Tomato mottle mosaic virus (ToMMV) (five sequences) and ToMV (10 sequences).

The obtained five primer pairs were tested by a real-time RT-PCR with SYBR green, in order to understand which primer pair has the lowest Ct value. The real-time RT-PCR was performed in a Q2plex HRM Platform Thermal Cycler (Qiagen, Hilden, Germany). The mixture consists in a 20 µl final volume containing one µl of total RNA extract (ToB-SIC01/19) with ∼10 ng RNA/µl concentration, one µl of one mM of each primer, 10 µl of Master mix QuantiNOVA XX (Qiagen, Hilden, Germany) and H2O Diethyl pyrocarbonate (DEPC) to reach the final volume.

Healthy tomato plant RNA and water were used as control samples. Each sample was analysed twice. Cycling conditions included reverse transcription at 48 °C for 10 min, incubation at 95 °C for 5 min, 45 cycles of 95 °C for 2 s and 60 °C for 20 s and fluorescence was measured at the end of each cycle. Melting curve steps were added at the end of RT-PCR as following: 95 °C for 1 min, 40 °C for 1 min, 70 °C for 1 min and a temperature increase to 95 °C at 0.5 °C/s to record the fluorescence.

A specific ToBRFV TaqMan® MGB probe (Eurofins Genomics, Ebersberg, Germany) was designed in a conserved domain within the region encompassed by the primers. The probe, with a length of 22 nucleotides, was 5′-labelled with the reporter dye FAM (6-carboxyfluorescein) and 3′-labelled with a minor groove binder non-fluorescent quencher (MGB NFQ). TaqMan MGB probes include a MGB moiety at the 3′ end, which increases the melting temperature (Tm) of the probe and stabilises the probe–target hybrids. Consequently, MGB probes can be significantly shorter than traditional probes, providing better sequence discrimination and flexibility to accommodate more targets. The predicted Tm values for ToBRFV primers and probe were 59–60 °C and 67 °C, respectively, calculated with the prediction tool provided by Primer Express Software v3.0.1. (Thermo Fisher Scientific, Waltham, MA, USA). The sequences included in this study are reported in Table S1.

ToBRFV genotype-specific real-time RT-PCR assay with TaqMan MGB probe

The real-time RT-PCR assay with TaqMan MGB probe was performed in a Rotor-Gene Q2plex HRM Platform Thermal Cycler (Qiagen, Hilden, Germany) in a reaction mix of 12 µl final volume, containing one µl of total RNA extract with the concentration of ∼10 ng RNA/µl, 0.5 µM of the forward primer ToB5520F and the reverse primer ToB5598R (the primer set that yielded the most sensitive detection for all ToBRFV isolates in any type of tissue providing the lowest Ct values), 0.25 μM of TaqMan MGB probe, 0.5 μl of RNase Inhibitor (Applied Biosystems, Foster City, CA, USA), six µl of 2× QuantiNova Probe RT-PCR Master Mix, 0.2 µl of QN Probe RT-Mix and H2O DEPC water to reach final volume.

The total RNA extracts used to perform this assay were obtained from five plants infected with ToBRFV and from eight tomato plants infected with Cucumber green mottle virus (CGMV), PaMMV, PMMV, TMGMV, TMV, ToMMV, ToMV and Zucchini green mottle mosaic virus (ZGMMV).

Each sample was analysed in duplicate in two independent real-time RT-PCR assays. The control samples in each run included total RNA from a healthy tomato plant, water instead of sample and at least two RNA transcript dilutions of the standard curve (see below). The probe annealed specifically in an internal region of the PCR product amplified with primers ToB5520F-ToB5598R. After binding, the probe is cleaved by the 5′ exonuclease activity of the DNA polymerase, which releases the reporter molecule away from the quencher, allowing the reporter dye to emit its characteristic fluorescence. The TaqMan MGB probes incorporate an NFQ to absorb the (quench) signal from the fluorescent dye label at the 3′ end of the probe.

The properties of the NFQ combined with the length of the MGB probe result in lower background signal than with no-MGB NFQ probes. Lower background means increased sensitivity and precision. The cycling conditions consisted in reverse transcription at 45 °C for 10 min, enzyme denaturation at 95 °C for 10 min, and 45 cycles of 95 °C for 5 s and 60 °C for 60 s with fluorescence measured at the end of each cycle. The mean (X) Ct value and the standard deviation for each tomato sample were calculated from the four Ct obtained values.

Standard curve

An external standard curve was generated to determine the sensitivity of the real-time RT-PCR protocol with the TaqMan MGB probe. Serial dilutions of an in vitro synthesised positive-sense RNA transcript of the selected gRNA region were amplified using the real-time RT-PCR TaqMan MGB assay. The template for the in vitro transcription was obtained by conventional RT-PCR amplification using total RNA extract from a tomato petiole infected with the just characterised ToBRFV ToB-SIC01/19 isolate. The obtained DNA product was cloned in the commercial pGEM-T vector, linearised with SalI enzyme and transcribed in vitro with the T7 RNA polymerase (New England Biolabs, Ipswich, MA, USA) following the manufacturer’s instructions.

Transcripts were purified with RNaid Spin kit (Bio101, CA, USA), treated twice with RNase free DNase (Turbo DNA-free from Ambion) and their concentration was determined in duplicate with NanoDrop 1000 spectrophotometer (Thermo Fisher Scientific, Waltham, MA, USA).

Ten-fold serial dilutions of the transcript in healthy tomato tRNA extract (10 ng/μl) containing 1010 to 101 copies were used in the real-time RT-PCR TaqMan MGB probe assay. The assay was performed with and without the reverse transcriptase, to ensure the absence of DNA template in transcript preparations.

The RNA transcript concentration (pmol) in each dilution was calculated with the formula: micrograms of transcript RNA × (106 pg/1 μg) × (1 pmol/340 pg) × (1/number of bases of the transcript), and the number of RNA copies was calculated using this concentration value and Avogadro’s constant. The standard curve was obtained plotting the threshold cycle (Ct) values from two independent assays with four replicates per standard dilution vs the logarithm of the RNA concentration dilution. The amplification efficiency was calculated from the slope of the corresponding curve using the formula 10(−1/slope of the standard curve), or the same formula × 100 (when given as a percentage value).

Different methods for sample preparation and comparison of different ToBRFV detection techniques

Four different procedures of sample preparation were evaluated using 40 samples from the 360 samples that were analysed previously by end-point RT-PCR (10 per province), the just characterised isolate ToB-SIC01/19 and a negative-tomato control plant. The obtained results were also compared by DAS-ELISA and end-point RT-PCR.

Immunocapture in real-time PCR tubes

Multiwell plates for real-time PCR were incubated at 37 °C for 1 h with 100 μl of polyclonal antibody for TMV (AGDIA, Elkhart, IN, USA) diluted 1:200 in a coating buffer (sodium carbonate anhydrous 1.59 g, sodium bicarbonate 2.93 g, sodium azide 0.2 g in one L of distilled water, pH 9.6). As reported by manufacture’s protocol, this antibody reacts with a variety of viruses from the Tobamovirus genus, such as Cucumber green mild mottle virus, Kyuri green mottle mosaic virus, PMMoV, TMV, ToBRFV and ToMV. After incubation, three washing steps were performed. A total of 100 μl of sap extract were obtained grinding the petioles in extraction buffer (sodium sulphite anhydrous 1.3 g, polyvinylpyrrolidone MW 24-40,000 20 g, powdered egg (chicken) albumin, Grade II 2 g, Tween-20 20 g in one L of distilled water, pH 7.4). After 1 h of incubation at room temperature, the multiwell was washed with standard washing buffer, dried and prepared for subsequent analysis.

Total RNA extraction

Total RNA (RNAt) was extracted from 0.1 g of the petiole, with the RNeasy Plant Mini Kit (Qiagen, Hilden, Germany) following the manufacturer’s instruction. RNA extracts were re-suspended in 30 μl of RNase-free water and adjusted to approximately 10 ng/μl using a NanoDrop 1000 spectrophotometer (Thermo Fisher Scientific, Waltham, MA, USA).

Direct crude extract

A slice of 0.4 mm of the petiole of each sample was directly placed in a 1.5 ml tube containing 0.5 ml of Glycine buffer (EDTA 1 mM, NaCl 0.05 M, Glycine 0.1M), vortexed for 30 s and heated at 95 °C for 10 min. Three microliters were used for subsequent analysis.

Leaf-disk crude extract

Five fresh-cut petioles were impressed in a one cm2 of Hybond®-N+ hybridisation membrane (GE Healthcare, Chicago, IL, USA), dried at room temperature for 5 min and placed in a 1.5 ml tube containing 0.5 ml of glycine buffer. Tubes were vortexed for 30 s and heated at 95 °C for 10 min. Three µl were used for the subsequent steps.

Finally, the 42 sample preparations were tested by DAS-ELISA using a commercial kit of polyclonal antibodies for TMV, which can also detect ToBRFV (AGDIA, Elkhart, IN, USA). The same samples were also tested by end-point RT-PCR using the primers ToBRFV-F-5722 and ToBRFV-R-6179, and by a real-time RT-PCR-MGB-probe based-method according to the protocols described previously.

In order to understand if the samples with high Ct value were really positive or false positives, the products obtained with the samples named 1A, 3C, 7R, 8R and 9R using the methods No. 3 and No. 4 of sample preparation were analysed in 2% agarose gel. The products were purified with the Ultraclean 96 PCR Cleanup Kit (Qiagen, Hilden, Germany) according to the manufacture’s instruction and sequenced in both directions using an ABI PRISM 3100 DNA sequence analyser (Applied Biosystems, Foster City, CA, USA).

Results

Screening of ToBRFV using end-point RT-PCR

The end-point RT-PCR analysis showed that 129 of the 360 analysed tomato samples (35.83%) were positive for ToBRFV. Table 1 reports the number of infected plants per province and data collection. As reported in Table 1, the incidence of ToBRFV was higher in the greenhouses in the provinces of Ragusa and Siracusa, with an infection percentage of 93.3% and 70%, respectively, than in the greenhouses of the provinces of Agrigento and Caltanissetta that showed a percentage of infected plants of 16.6% and 6.66%, respectively. Analysing the three surveys performed in the four provinces, the virus showed a downward trend. Indeed, Ragusa decreased from 93.3% of the first sampling to 63.3% of the third sampling, Siracusa from 70% to 40% and Agrigento and Caltanissetta passed from 16.6% to 6.66%, respectively, to zero.

Table 1 Number of ToBRFV-infected tomato plants detected by endpoint RT-PCR.

Province	No. ToBRFV-infected plants/No. collected plants	
2018-October	2018-December	2019-February	
Agrigento	2/30	0/30	0/30	
Caltanissetta	5/30	0/30	0/30	
Ragusa	28/30	22/30	19/30	
Siracusa	21/30	20/30	12/30	

Mechanical transmission

Mechanical inoculations successfully transmitted ToB-SIC01/19 in all the plants of S. lycopersicum and C. annuum, while no transmission occurred in S. melongena.

The tomato-inoculated plants did not show any symptom until 22 dpi. Starting from the day 22, tomato plants showed symptoms that consist in interveinal yellowing, deformation and mosaic in young leaves. In pepper-inoculated plants, the symptoms started at 23 dpi and consisted in slight interveinal yellowing on young leaves and necrosis on the stem. RT-PCR performed at 30 dpi confirmed the presence of ToBRFV in tomato and pepper plants and its absence in eggplants (Table 2).

Table 2 Mechanical inoculations and symptoms caused by ToB-SIC01/19 isolate on different herbaceous species.

Species	Time after inoculaiton (weeks)	RT-PCR	% infected plants	
1	2	3	4	5	
Solanum lycopersicum	−	−	−	iy, myl	iy, myl, dyl	+	100%	
Capsicum annum	−	−	−	siy	siyl, ns	+	100%	
Solanum melongena	−	−	−	−	−	−	−	
Note:

iy, interveinal yellowing; myl, mosaic in young leaves; dyl, deformation in young leaves; siyl, slight interveinal yellowing in young leaves; ns, necrosis in the stem.

Full genome sequencing

The gRNA of the ToBRFV isolate named ToB-SIC01/19 of Sicily was completely sequenced in this study. An RT-PCR synthesis was designed using the genome walking strategy of overlapping fragments for each adjacent amplified product to avoid the sequencing of different templates (Luria et al., 2017). The accuracy of pure isolate sequencing was assured by the nucleotide sequences comparison of five different clones of each amplicon. The ToB-SIC01/19 sequence was assembled with the programme Contig implemented with the Vector NTI Advance 11.5 software (Invitrogen, Carlsbad, CA, USA) and deposited in GenBank under the Acc. No. MN167466.

The genome length of ToB-SIC01/19 consists in 6,391 nucleotides and it was organised as just reported for ToBRFV (Luria et al., 2017). ToB-SIC01/19 showed a percentage identity of 99.8%, 99.7%, 99.7% and 99.7% with the sequences of TBRFV-P12-3H (Germany; Acc. No. MK133095), TBRFV-P12-3G (Germany; Acc. No. MK133093), Tom1-Jo (Jordan, Acc. No KT383474) and TBRFV-IL (Israel, Acc. No KX619418), respectively.

Primer, Taqman® MGB probe design and protocol optimisation

The in vitro analysis using Primer-BLAST algorithm of the five primers pairs designed to target the MP gene showed no relevant match with other organisms. All the obtained primer pairs were tested by the real-time RT-PCR with SYBR green to identify the primer pair with the lowest Ct value. Table 3 reports the five designed primer pairs. Between the five tested primer pairs, two did not give any signal (ToB5461F/ToB5592R and ToB5461F/ToB5593R), while the primer pair ToB5520F/ToB5598R showed the lowest Ct value and was used for the subsequent analyses (Figs. 2A and 2B).

Figure 2 Real time RT-PCR using the SYBR green method with different primer pairs.

(A) Amplification curves of real time—RT-PCR using SYBR green with the following primer pairs: p1 (ToB5520F/ToB5598R), p2 (ToB5498F/ToB5683R), p3 (ToB5461F/ToB5683R), p4 (ToB5461F/ToB5592R) and p5 (ToB5461F/ToB5593R). Primer pairs p4 and p5 did not show amplification, while primer pair A showed the lowest Ct value. (B) Melting curves of the amplification curves previously obtained with the five different primer pairs.

Table 3 Forward, reverse primers and probe designed for quantitative RT-qPCR for the detection of Tomato brown rugose fruit virus (ToBRFV).

Name	Genomic
position	Referring
sequence	Sequence (5′-3′)	Amplicon
Size (bp)	Ct
value	
ToB-probe	5558	KT383474	FAM-GTTTAGTAGTAAAAGTGAGAAT-MGB			
ToB5520F	5520	GTAAGGCTTGCAAAATTTCGTTCG	101	5.0	
ToB5598R	5598	CTTTGGTTTTTGTCTGGTTTCGG	
ToB5498F	5498	CATGGAAGAAGTCCCGATGT	205	20.8	
ToB5683R	5683	ATCTGAATCGGCGACGTAAG	
ToB5461F	5461	GGCCCATGGAACTATCAGAA	242	25.6	
ToB5683R	5683	ATCTGAATCGGCGACGTAAG	
ToB5461F	5461	GGCCCATGGAACTATCAGAA	151	n.a.	
ToB5592R	5592	TTGTCTGGTTTCGGCCTATT	
ToB5461F	5461	GGCCCATGGAACTATCAGAA	152	n.a.	
ToB5593R	5593	TTTGTCTGGTTTCGGCCTAT	
Note:

n.a., not amplification.

In vitro hybridisation analysis of the primer pair ToB5520F and ToB5598R against other tobamoviruses was carried out with the programme Vector NTI 11.5 programme (Invitrogen, Carlsbad, CA, USA) and results showed that no relevant matches were identified (see Table S1).

ToBRFV specific real-time RT-PCR assay with TaqMan MGB probe

To determine the specificity of the real-time RT-PCR assay with TaqMan MGB probe, 17 different samples were analysed. As reported in Table 4, the two RNA transcripts gave the most sensitive signal with a Ct value ranging from 5.0 to 5.1 in four different assays. The total RNA derived from artificially ToBRFV-infected plants gave also positive signal with a Ct value that ranged from 12.2 ± 0.2 to 18.1 ± 0.1, while the other tobamoviruses used as outgroups did not give any signal.

Table 4 Real time RT-PCR assay with TaqMan MGB probe.

Comparison between ToBRFV and other tobamovirus.

Virus isolate	Ct ± SD	
ToB-SIC1/19-P1	18.1 ± 0.1	
ToB-SIC1/19-P2	14.2 ± 0.3	
ToB-SIC1/19-P3	12.2 ± 0.2	
ToB-SIC1/19-P4	13.5 ± 0.2	
ToB-SIC1/19-P5	12.9 ± 0.1	
Healthy tomato plant	–	
tRNA1	5.1 ± 0.0	
tRNA2	5.0 ± 0.0	
CGMV	–	
PaMMV	–	
PMMV	–	
TMGMV	–	
TMV	–	
ToMMV	–	
ToMV	–	
ZGMMV	–	

Standard curve

In order to calculate the number of RNA copies and the sensibility threshold of the developed technique, a standard curve was generated using 10-fold serial dilutions of RNA in vitro transcripts (from 1010 to 101 copies) of ToBRFV ToB-SIC01/19 isolate in healthy tomato RNAt. The standard curve covered a wide dynamic range (10 units of concentration) and showed a strong linear relationship, with a correlation coefficient of 0.9997 and 100% amplification efficiency (Figs. 3A and 3B). The real-time RT-PCR assay and the standard curve enabled the detection of as few as 101 ToBRFV RNA copies in tomato extracts and were used to determine the number of RNA copies in the total RNA extracts from the tomato samples collected in four different areas of Sicily, within the provinces of Agrigento, Caltanissetta, Ragusa and Siracusa. The average Ct values were within the dynamic range of the standard curve and ranged from 14 to 23.

Figure 3 Standard curve and linear regression of real time RT-PCR.

(A) Standard curves prepared with 10-fold serial dilutions of in vitro-synthesised RNA transcripts from ToBRFV ToB-SIC01/19 isolate using real time RT-PCR with TaqMan MGB probe. (B) Curves were generated by linear regression analysis, plotting the Ct value in the Y-axis vs the logarithm of the starting RNA dilutions in the X-axis. Each plotted point represents the mean Ct value that was calculated from the four different experiments with two replicates. The calculated correlation coefficient (R2) and amplification efficiency (E) values are indicated in each curve.

Different methods for sample preparation and comparison of different ToBRFV detection techniques

Four different procedures were evaluated to identify the best method for sample preparation. A total of 42 samples (10 per province, ToB-SIC01/19 and a healthy tomato plant) were included. The results obtained with the four different methods of sample preparation were compared with each other and with DAS-ELISA and end-point RT-PCR (Table 5). All the four different methods for sample preparation (immunocapture, total RNA extraction, direct crude extract and leaf-disk crude extract) were effective for ToBRFV detection by real-time RT-PCR. Analysing the four different procedures, the Ct value obtained with immunocapture ranged from 16 to 28, with total RNA extraction from 14 to 23, with leaf-disk crude extract from 17 to 35 and with direct crude extract from 17 to 37.

Table 5 Analysis of 42 samples using different methods of sample preparation and comparison of real time RT-PCR against DAS-ELISA and endpoint RT-PCR.

Province	Samples	DAS-ELISA	RT-PCR end point	Real time RT-PCR probe Ct value	
1*	2*	3*	4*	1*	2*	3*	4*	
Agrigento	1A	−	−	−	−	−	−	−	−	−	
2A	−	−	−	−	−	−	−	−	−	
3A	−	−	−	−	−	−	−	−	−	
4A	−	−	−	−	−	−	−	−	−	
5A	−	−	−	−	−	−	−	−	−	
6A	−	−	−	−	−	−	−	−	−	
7A	−	−	−	−	−	−	−	−	−	
8A	−	−	−	−	−	−	−	−	−	
9A	−	+	+	−	−	18	18	26	26	
10A	−	+	+	−	−	21	19	27	28	
Caltanissetta	1C	−	−	−	−	−	−	−	−	−	
2C	−	−	−	−	−	−	−	−	−	
3C	−	−	−	−	−	−	−	−	−	
4C	−	−	−	−	−	−	−	−	−	
5C	−	+	+	−	−	20	17	30	30	
6C	−	+	+	−	−	20	17	25	25	
7C	+	+	+	+	+	22	18	31	32	
8C	+	+	+	−	−	22	19	31	33	
9C	−	−	−	−	−	−	−	−	−	
10C	−	+	+	−	−	24	21	32	32	
Ragusa	1R	+	+	+	+	−	22	18	30	25	
2R	+	+	+	+	−	22	18	28	29	
3R	−	+	+	−	−	26	21	32	32	
4R	−	+	+	−	−	26	19	33	34	
5R	−	−	−	−	−	−	−	−	−	
6R	+	+	+	+	+	26	17	31	27	
7R	+	+	+	+	+	24	17	30	32	
8R	−	+	+	−	−	28	19	33	34	
9R	−	+	+	−	−	28	21	35	37	
10R	+	+	+	−	−	26	19	32	31	
Siracusa	1S	+	+	+	−	−	26	19	32	32	
2S	+	+	+	−	−	22	18	30	29	
3S	+	+	+	−	−	22	17	31	31	
4S	−	+	+	−	−	26	21	32	33	
5S	−	+	+	−	−	26	23	34	34	
6S	+	+	+	+	−	23	19	32	32	
7S	+	+	+	+	+	20	17	29	29	
8S	−	−	−	−	−	−	−	−	−	
9S	−	−	−	−	−	−	−	−	−	
10S	−	−	−	−	−	−	−	−	−	
	ToB-SIC1/19	+	+	+	−	−	16	14	17	17	
	Healthy plant	−	−	−	−	−	−	−	−	−	
Note:

1* Immunocapture in RT-qPCR multiwell; 2* Total RNA extraction; 3* Leaf-disk crude extract; 4* Direct crude extract.

Regarding the comparison between the real-time RT-PCR and other techniques, the real-time RT-PCR showed more sensitivity than DAS-ELISA; indeed, 11 samples that gave negative results by DAS-ELISA were positive with the real-time RT-PCR (Table 5). Comparing the real-time RT-PCR to the end-point RT-PCR, the two techniques gave the same results with immunocapture and total RNA extraction procedure, while using direct crude extract and leaf-disk crude extract the end-point RT-PCR was unfit to provide a reliable result.

To confirm the absence of false positives in the real time RT-PCR assay, the amplification products of the samples named 1A, 3C, 7R, 8R and 9R were analysed in 2% agarose gel. The samples 7R, 8R and 9R gave the expected fragment while the samples 1A, 3C, healthy plant and H2O did not show amplification (Fig. 4). Sequencing of the samples 7R, 8R and 9R with the sample preparation methods No. 3 and No. 4 confirmed the presence of ToBRFV.

Figure 4 Comparison of samples showing high Ct value in real timeRT-PCR with electrophoretic gel.

(A) Amplification curves obtained with the primer pair ToB5520F/ToB5598R and ToBprobe-5558 probe of the samples reported in Table 5, which are named: tRNA1, 1A, 3C, HP (Healthy Plant), H2O, 7R, 8R and 9R. The 3* and 4* indicated the different samples preparation methods that are reported in Table 5. (B) Electrophoretic 2% agarose gel of real-time RT-PCR products. M: 100 bp marker (Thermo Fischer Scientific, Waltham, MA, USA).

Discussion

Sicily is one of the Mediterranean Basin regions with the most important tomato production and, due to its geographical position, it represents the main access point for plant material to the European countries. This situation considerably increases the risk of introducing new pathogens into our environments and entails a serious risk for agriculture biosecurity and food production, jeopardising the future of Italian horticulture. The outbreak of ToBRFV represents a threat due to its multiple transmission methods and to the absence of tomato and peppers resistant varieties. Furthermore, several hybrid tomato varieties presenting ToMV and TMV Tm-1, Tm-2 and Tm-22 resistance genes (Pelham, 1966), can be severely affected by ToBRFV, leading to a rapid virus spread in all those areas where tomato is cultivated. To date, the only two available tools to contain ToBRFV worldwide are early diagnosis and the implementation of preventive measures in crop management, which can be a valuable aid in reducing the introduction and subsequent ToBRFV spread in other countries. Consequently, today there is the need to develop alternative, sensitive and very reliable diagnostic methods for the diagnosis of plant viruses, which are compromising the tomato crops in various Italian and international areas (Hanssen, Lapidot & Thomma, 2010; Puchades et al., 2017; Ferriol et al., 2015, Panno et al., 2014). In the present work a ToBRFV isolate, recently found in Sicily (Panno, Caruso & Davino, 2019), was characterised, and a sensitive, specific, rapid and economical method for its detection in infected plants and seeds was developed.

The biological characterisation of the Sicilian ToB-SIC01/19 isolate demonstrated the possibility of mechanical transmission on tomato and pepper plants, as previously reported by Luria et al. (2017), while on eggplant it cannot be transmitted. Mechanical transmission is very important because it could simplify genetic improvement activities, in order to constitute new tolerant/resistant germplasm towards ToBRFV. Molecular characterisation showed that the ToBRFV Sicilian isolate ToB-SIC01/19 presents a percentage identity of 99% with the sequences retrieved in Germany, Jordan and Israel (Menzel et al., 2019; Salem et al., 2016; Luria et al., 2017). Since ToBRFV spread in a few years within the Mediterranean Basin countries and in Central America, the very low level of variability found among isolates supports the hypothesis that the recent introduction in Italy probably occurred through infected seeds.

The ability of the virus to transmit trough plant-to-plant contact, manipulation and especially by seeds in overlapping crops cycles, such as tomato intensive cultivation in the greenhouses, facilitated the rapid spread of the virus.

For these reasons, we have developed a quick detection procedure for ToBRFV diagnosis based on real-time RT-PCR TaqMan MGB probe. This method detects ToBRFV in samples obtained by different preparation procedures. Additionally, the direct crude extracts and leaf-disk crude extracts were successfully used to avoid total RNA extraction, shortening the processing time, allowing the simultaneous analysis of multiple samples and drastically reducing the total cost for single analysis. Its high sensitivity is relevant to minimise false negatives and to obtain correct discrimination of ToBRFV with other viruses belonging to the genus Tobamovirus. Furthermore, the developed technique, associated with the direct crude extract sample preparation, can be used to make in-field diagnosis with a portable device, allowing a considerable saving of time and it could also be used by non-technical personnel.

Conclusions

The method developed in this work is based on the use of a real-time RT-PCR TaqMan MGB probe and could represent a good and reliable tool to be included in certification programs. Currently, immuno-enzymatic methods, such as DAS-ELISA, are used in some cases for plant virus diagnosis, but these tests may not be very reliable and give ‘false negatives’, due to the low viral titre of nursery plants and early infections, and to the absence of specific antibodies (Jacobi et al., 1998).

Moreover, the method developed in the present study requires short time and allows the analysis of a great number of samples at the same time, when associated with the use of leaf-disk crude and direct crude extracts. For this reason, this method could be used as a routine test in the laboratories of vegetable diagnosis. In conclusion, to avoid ToBRFV spread to other Italian and European regions, phytosanitary actions are required, such as correct crop management, more restrictive measures at international borders using rapid, sensitive and economical tools for diagnosis and, more importantly, the development of tomato resistant cultivars that have the ability to tolerate the disease. To date, new tomato cultivars that are tolerant/resistant to ToBRFV are not available. For this reason, the only way to contain this disease globally is the use of a sensitive and economical diagnostic tool such as the Real time RT-PCR method that was developed and described in this paper.

Supplemental Information

Supplemental Information 1 Sequencing of Real time RT-PCR products.

Click here for additional data file.

Supplemental Information 2 RAW_DATA_Standard_curve.

Standard curves prepared with 10-fold serial dilutions of in vitro-synthesized RNA transcripts from ToBRFV isolate ToB-SIC01/19 using RT-qPCR with TaqMan MGB probe.

Click here for additional data file.

Supplemental Information 3 Hybridisation percentage (%) of primer ToB5520F, ToB5598R and ToB-probe5558 against different tobamoviruses that affect tomato and pepper. In the table are reported only the data with the highest sequence homology inside each species.

Click here for additional data file.

Additional Information and Declarations

Competing Interests

Author Contributions

DNA Deposition

Data Availability

Stefano Panno is employed by Molecular Dynamics Srl.

Stefano Panno performed the experiments, analysed the data, authored or reviewed drafts of the paper, approved the final draft.

Susana Ruiz-Ruiz performed the experiments, analysed the data, authored or reviewed drafts of the paper, approved the final draft.

Andrea Giovanni Caruso performed the experiments, analysed the data, prepared figures and/or tables, approved the final draft, bioinformatic analysis.

Ana Alfaro-Fernandez analysed the data, contributed reagents/materials/analysis tools, prepared figures and/or tables, approved the final draft.

Maria Isabel Font San Ambrosio analysed the data, contributed reagents/materials/analysis tools, prepared figures and/or tables, authored or reviewed drafts of the paper, approved the final draft.

Salvatore Davino conceived and designed the experiments, contributed reagents/materials/analysis tools, authored or reviewed drafts of the paper, approved the final draft.

The following information was supplied regarding the deposition of DNA sequences:

The sequence data is available at GenBank: MN167466.

The following information was supplied regarding data availability:

Raw data are available in the Supplemental Files.

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
