# Peer review of "Real-time reverse transcription polymerase chain reaction development for rapid detection of Tomato brown rugose fruit virus and comparison with other techniques"

_PeerJ, doi:10.7717/peerj.7928_

## Round 0.1 · original submission · Minor Revisions

Dear colleagues,

According to two reviewers, the manuscript is of broad interest, and has been performed and written adequately.

In my view:

1. As reviewer #2 pointed out, the written English needs a little more work. Some sentences are indeed a bit awkward, but it is not a massive issue. Please request a native speaking colleague to do a quick proof reading and fix the few strange sentences.

2. Both reviewers, but especially reviewer #2, point out that more detail needs to be given in reporting results. Please address each of these recommendations specifically, adding the data/tables that were requested.

·

Basic reporting

The manuscript was written in good English; but it can be improved.
The literature references were good enough to the topic
The manuscript was structured well and the authors expressed themselves in.

Experimental design

The experiments were carried ad hoc and the methods were sufficient

Validity of the findings

The findings are novel and useful. More needed to explain how these can be used as a routine tests. The recommendations need to be improved based on that

Additional comments

The manuscript is original and informative.

Reviewer 2 ·

Basic reporting

The article develops new qRT-PCR primers and probe to determine the presence of a tomato virus that recently affected the south of Italy. Because of the importance of this crop in this region I think that the possibility to detect this disease in a easier way is crucial. In my point of view, the most relevant and original section of this work is the fact that the author explored the possibility of manipulate in a lesser extend the samples. So, they offer an easy, cheap and sensitive way to detect the virus.
I consider very important to correct some aspects of this work previous of publishing.
In general the English writing must be check out in all the manuscript, I suggest to contact some expert in English writing.
Some examples of sentences that according with my point of view need to be corrected:
In Bacground: In Sicily, the first outbreak occurred in 2018 in different Sicilian provinces represents a serious threat for tomato crops.
Line 54: In Italy the tomato production is more located in Sicily, where is one of the most economically horticultural crops, with the yield of approximately 40% of the total Italian production in greenhouses (Agri ISTAT, 2017).
Line 67: instead of makes should say make.
Line 70: instead of subsequently should say subsequent.
Line 342: it should say sample, not samples.
Line 357: should say the most important tomato production, not productions
Check this sentence: So, with regard to this pathogen, to date the only two tools available are:
I think that the world cultivation is used in a wrong way in many sentences, please review this in all the text.

Experimental design

Some issues that on my point of view need to be address to improve the work:

In results, section Mechanical transmission, the authors said that they could infect with the virus tomato and pepper plants, but not eggplant. And they detected the virus by PCR. There is no reference to a Table or Figure where the results are shown. For example: percentage of the total infected plants with symptoms, positive or negative PCR result. I suggest perform a graph showing these results.

In results, section Primer, Taqman® MGB probe design and protocol optimization, the author claim that their designed 5 pairs of primers for qRT-PCR and showed the one with the highest sensibility in Table 2. What means highest sensibility? Lower Cq values? Earlier virus detection? Better primer efficiency. I think that if the section is named protocol optimization they should show all the primers sequences with their corresponding primer efficiency and R2 values. Also, they should show the parameter that they measured that helped them to decide for the best one. Also, I think that it is very important when a person optimize a qRT-PCR show the melting curves in a supplementary figure.

Supplementary Table 1 needs more information. It is not clear what means % or at what correspond the Ac. numbers, I mean if they are all of the analyzed. It is not an intuitive table and there is no extra information in Results.

In section Different method for sample preparation and comparison of different techniques
for ToBRFV detection:
I think this is the most interesting result of the paper, the comparison between different isolation techniques and virus determination in my opinion is the most applicable part of this paper. But, my concern is about the final conclusion. The author said that the qRT-PCR method is the most sensitive because can detect the virus even with crude tissue extracts. However, most of the Cqs obtained using this extracts as template are higher than 30. These high Cq in most of the cases are considered as noise. I think that it is necessary show that these are real viral amplifications showing for example qRT-PCR negative controls.

Material and Methods:
Line 117: Why do you use 1 mM of primer F and 1 uM primer reverse? Is this a mistake?

There is an excel file (supplementary information) that is not mentioned in the paper at all.

Validity of the findings

As I mentioned before, I think that the author have developed a novel way of virus examination for tomato samples. Compared with the usually used techniques in the field seems to be a cheaper and easy way to do it. That's the novelty of the paper.

However and as I mention in the previous sections of this revision, I think that several issues need to be check out before to publish this work

Additional comments

I suggest a deeper revision of the English writing. Some sentences are not clear and not easy to follow.

Please, check out the tables, some of them need more details for a better interpretation of the results.

Please, add the information suggested before. Some results affirmations are not supported with data.

---

## Round 0.2 · accepted · Accept

Thank you for addressing and implementing all reviewers comments. The writing as well has been much improved after correction by an English expert.

Congratulations on the very nice article!